# Base Editing: The Ever Expanding Clustered Regularly Interspaced Short Palindromic Repeats (CRISPR) Tool Kit for Precise Genome Editing in Plants

**DOI:** 10.3390/genes11040466

**Published:** 2020-04-24

**Authors:** Mahmuda Binte Monsur, Gaoneng Shao, Yusong Lv, Shakeel Ahmad, Xiangjin Wei, Peisong Hu, Shaoqing Tang

**Affiliations:** State Key Laboratory of Rice Biology and China National Center for Rice Improvement, China National Rice Research Institute, Hangzhou 310006, China; mahmudamumu0@gmail.com (M.B.M.); shaogaoneng@caas.cn (G.S.); lvyusong0907@163.com (Y.L.); shakeelpbg@gmail.com (S.A.); weixiangjin@caas.cn (X.W.); hupeisong@caas.cn (P.H.)

**Keywords:** clustered regularly interspaced short palindromic repeat, base editor, point mutation, genome editing, base conversion

## Abstract

Clustered regularly interspaced short palindromic repeats (CRISPR)/CRISPR associated protein 9 (Cas9), a newly developed genome-editing tool, has revolutionized animal and plant genetics by facilitating modification of target genes. This simple, convenient base-editing technology was developed to improve the precision of genome editing. Base editors generate precise point mutations by permanent base conversion at a specific point, with very low levels of insertions and deletions. Different plant base editors have been established by fusing various nucleobase deaminases with Cas9, Cas13, or Cas12a (Cpf1), proteins. Adenine base editors can efficiently convert adenine (A) to guanine (G), whereas cytosine base editors can convert cytosine (C) to thymine (T) in the target region. RNA base editors can induce a base substitution of A to inosine (I) or C to uracil (U). In this review, we describe the precision of base editing systems and their revolutionary applications in plant science; we also discuss the limitations and future perspectives of this approach.

## 1. Introduction

The development of genome-editing technology has revolutionized agricultural studies, enabling the improvement of different plant traits [1,2,3,4,5]. Various tools have been widely used for precise genome editing in plants, including clustered regularly interspaced short palindromic repeats (CRISPR) based gene editing techniques, such as CRISPR/CRISPR associated protein 9 (Cas9), CRISPR/Cas12a (Cpf1), and CRISPR/Cas13 [6,7,8,9], owing to their reliability. All these techniques have certain advantages and disadvantages and have become more precise, efficient, and robust in recent years [1,10,11,12,13]. The formation of DNA double-strand breaks (DSBs) at target loci is one of the primary features of CRISPR/Cas genome editing. Different outcomes can be achieved via two major DNA repair pathways, i.e., nonhomologous end joining (NHEJ) and homology-directed repair (HDR) pathways. Gene insertion, replacement, and insertion/deletion (indel) mutations are outcomes of the NHEJ pathway, which does not require any homologous repair template [7]. In contrast, by adopting an available exogenous DNA donor template, gene replacements, insertions, and corrections can be achieved via the HDR pathway [8]. NHEJ is efficient and widely used for single/multiplex editing and large-scale mutations. However, its precision remains questionable in terms of point mutation or single base conversion [2,10,14]. Therefore, to ensure effective point mutations in a target gene, HDR-mediated genome-editing technique has been widely applied in different organisms. Despite this, its application is somehow limited in plants owing to the limited availability of donor templates in plant cells [7,15,16]. However, recent development of potential CRISPR/Cas9 gene targeting technology found helpful for homozygous recombination in plants [17]. Moreover, Ali et al. [18] combine *Agrobacterium* VirD2 relaxase with Cas9 endonuclease and generate potential Cas9-VirD2 for allele modification in rice.

To overcome these limitations, researchers have aimed to develop editing technologies with an improved efficiency, reliability, and precision. Base editing is a newly developed technique for precise genome editing that enables irreversible base conversion at a specific site. The base editor (BE) is a complex of a catalytically impaired Cas protein, guide RNA (gRNA), and a nucleobase deaminase domain which is able to convert specific base pairs [8,19,20,21]. Base editing is simple and precise for nucleotide conversions that do not involve the formation of DNA DSBs [22,23]. The cytosine BE (CBE), which was the first developed type, shows a high efficiency in the conversion of cytosine (C) to thymine (T) [24,25,26]. As understanding of the molecular functions of deaminases increased, another base editing tool, the adenine BE (ABE), was developed for the efficient conversion of adenine (A) to guanine (G) [24,27]. Using the Cas13 technology, an RNA BE (RBE) has also emerged as a base editing tool, able to convert A to inosine (I) in RNA [28,29]. RNA editing for specific C to uracil (U) exchange (RESCUE) has also been developed as an efficient tool [30]. BEs are further developed to overcome the limitations of other genome-editing tools and are expected to have wide applications in many fields in the future.

In this review, we discuss the development of various plant BEs (PBEs), focusing on comparing the precision of base-editing tools with that of other CRISPR tools and examining their increasing applications in crop improvement within a short period. Furthermore, future perspectives and current limitations of base-editing tools are discussed.

## 2. Evolution of Base Editors

CRISPR/Cas9 was first described in 2002, although the CRISPR mechanism was discovered in 1987 [31,32]. Cas9 from *Streptococcus pyogenes* (SpCas9), which required a protospacer adjacent motif (PAM) sequence, NGG, was most commonly used. To overcome the G-rich PAM preference, Cpf1 (also known as Cas12a) was developed in 2015 for targeting T-rich PAMs [33,34]. In 2017, the ADAR mediated RBE technology was reported for A to I conversion [28] (Figure 1).

The CRISPR/Cas9 and Cpf1 tools are widely used for different purposes; however, their application for precise and sophisticated genome editing remains limited. Targeted precise point mutations or single-base conversions (base editing) remain difficult to achieve. Base editing or a specific base conversion enables the replacement of a target base pair with a different base pair in a desirable manner, without inducing DNA DSBs [9,22]. In contrast to DSB–HDR-mediated genome editing, site-specific substitutions of DNA bases result in fewer indels [35]. CRISPR/Cas9 mainly relies on homologous recombination [36,37,38,39]. In addition, base editing is more specific than the CRISPR/Cas9 technology because the rate of off-target mutations is lower in adenine base editing than the former [7,40,41]. Previous studies have indicated that CBEs contain more off-target mutations than ABEs [40,42]. However, recent studies showed that this unwanted effect of CBEs can be strongly limited [43,44]. Tan et al. [43] developed precise CBEs with low off-target mutation by engineering CDA1 truncations and nCas9 fusions. Moreover, Doman et al. [44] minimized Cas9 independent off-target DNA editing by engineered CBEs. Thus, base editing has overcome the limitations of previous CRISPR-based technologies.

BEs restrict the generation of indels at both target and off-target sites, as there is no requirement for DNA modification via DSBs [23,25,45,46], and play important roles in single-base conversions or nucleotide substitutions, independent of the DNA donor [23]. Several base editing systems have been developed with different goals. BEs are mainly categorized as CBEs which can convert C to T, ABEs which convert A to G, and RBEs which convert A to I or C to U. CBEs mainly comprise four modules, including single guide RNA (sgRNA), nCas9 (D10A), or dCas9, a cytosine deaminase, and a uracil DNA glycosylase inhibitor (UGI) (Figure 2a). The function of sgRNA is to guide a Cas9-cytosine deaminase fusion to bind to the target sequence, after which the cytosine deaminase catalyzes the base conversion [47]. UGI is an 83-residue protein from *Bacillus subtilis* bacteriophage PBS1 which blocks the human uracil DNA glycosylase (UDG) activity. The Cas9 nickase (nCas9) and catalytically inactive Cas9 (dCas9) are derived from a wild-type Cas9, with one (nCas9) and two (dCas9) amino acid mutations, respectively. Komor et al. [25] reported a first generation BE (BE1), composed of the rat APOBEC1/XTEN/dCas9 proteins. However, the C to T substitution efficiency using BE1 was only 0.8–7.7% of total DNA sequences because UDG catalyzed the removal of U from DNA, creating C:G pairs instead of U:G pairs. The second-generation BE (BE2) was created by fusing UGI to the C terminus of BE1 (APOBEC/XTEN/dCas9/UGI), and the third generation BE (BE3) was created by replacing dCas9 with nCas9 (D10A). Additionally, rAPOBEC1 was fused to the N terminus of nCas9 D10A, through an XTEN linker to increase the editing efficiency of BEs. At specific target sites, UGI obstructs endogenous base excision repair [39].

Different types of Cas proteins and engineered Cas9 proteins have been shown to effectively recognize different PAMs and can be utilized as BEs [48]. Several studies have successfully expanded the PAM specificity of BEs using different engineered Cas9 proteins; these systems are not limited by the presence of NGG PAMs but can also recognize NG, AGG, GAA, GAG, CAG, GAT, NGA, and NGCG PAMs [42,49,50]. Kleinstiver et al. [51] introduced mutations into SpCas9 and altered the PAM specificity of CRISPR/Cas9. Additionally, Kim et al. [41] used mutated SpCas9 proteins and produced a series of BEs (VQR-BE3, EQR-BE3, VRER-BE3, and SaKKH-BE3) that could target NGAN, NGAG, NGCG, and NNNRRT PAMs, respectively. These authors also replaced SpCas9 in the BE3 with SaCas9 to generate APOBEC1/SaCas9n/UGI (SaBE3), which showed a higher editing efficiency and a broader PAM specificity. Based on *Staphylococcus aureus* Cas9 (SaCas9) and SaKKH, Qin et al. [52] developed series of ABEs and CBEs. PAM specificity of SaCas9 was relaxed to NNGRRT by SaCas9-KKH. These editing tools successfully expanded the scope of genome editing to target with a NNGRRT PAM. In plants *Petromyzon marinus* cytidine deaminase 1 (PmCDA1) is also efficient as rAPOBEC1 based CBEs [53]. Wu et al. [54] expanded base editing in rice by PmCDA1/SpCas9/UGI with 4–90% mutation efficiency. Moreover, Xu et al. [55] reported high efficiency of PmCDA1 than rAPOBEC1 in rice suggesting more scope of PmCDA1 based CBEs. Efficient single nucleotide substitution was successfully achieved by engineering the fusion between the Cas domain of the editor and the deaminase domain [15,56]. This CBE tool could also be used in plants. Moreover, a recent study has reported base editing in human cells using Cpf1 cytidine deaminase fusion [57]. Using this approach, the limitation of only selecting targets with G/C rich PAM sequences was overcome because the Cpf1 based CBE can efficiently convert C to T and recognize T-rich PAM sequences. Cpf1 uses only a single nuclease domain to cleave the NT-strand and T-strand sequentially. Thus it is difficult to engineer Cpf1 into a nickase that solely cleaves T-strand as nCas9 in BE3 does. Consequently, researchers fused a catalytically inactive *Lachnospiraceae bacterium* Cpf1 (dLbCpf1) or *Acidaminococcus* sp. Cpf1 (dAsCpf1) with rAPOBEC1 and UGI, thus creating dCpf1-BEs, dLbCpf1-BE0 and dAsCpf1-BE0. With further modifications, a series of Cpf1-BEs were developed to improve the editing efficiency.

ABEs were developed later for converting A to G. Unlike in CBEs, there is no need for a DNA glycosylase inhibitor in ABEs, and natural adenine deaminase cannot accept DNA. Liu’s group developed several generations of ABEs. Transfer RNA adenosine deaminase (TadA) from *Escherichia coli* was mutated to obtain a deoxyadenosine deaminase (TadA*), and TadA-TadA* heterodimer was generated via fusion [27]. ABEs were then developed (Figure 2b) by linking this heterodimer to nCas9 (D10A) and seventh-generation ABEs (ABE7.10) effectively convert A to G. Replacement of the SV40 nuclear localization signal (NLS) in ABE7.10 with bis-bpNLS increased the editing efficiency. Additionally, codon optimization in bis-bpNLS ABE7.10 (ABEmax) resulted in an increased editing efficiency compared with that of ABE7.10 [58]. Hu et al. [59] replaced SpCas9 in ABE 7.10 with xCas9-3.7 because specificity of xCas9 is not limited to NGG PAMs. The xCas9 variant of SpCas9 could target NG, GAA, and GAT PAMs, which successfully expanded the scope of genome editing in rice [59,60]. In plants and human cells, SpCas9-NG variant was successfully tested that recognizes the NGN PAM [61,62,63,64]. Veillet et al. [63] compared the efficiency of xCas9, SpCas9, and SpCas9-NG and found SpCas9-NG targets more efficiently alternative NGT PAMs in plants.

The CRISPR/Cas effector Cas13a (previously known as C2c2) was engineered for RNA knockdown and binding. An RBE system was developed through the RNA editing for programmable A-to-I replacement (REPAIR) system using a combination of a catalytically inactive Cas13 (dCas13) and an adenosine deaminase acting on RNA (ADAR) [29]. A Cas13b ortholog from *Prevotella* sp. P5-125 (PspCas13b) was found to be highly specific, and generation of REPAIRv2 via dCas13b–ADAR2DD fusion resulted in a highly specific RNA-editing tool (Figure 2c) [15,29,65]. Qu et al. [66] developed a leveraging endogenous ADAR for programmable editing of RNA systems using short engineered ADAR-recruiting RNAs to alter specific A bases to I. RBEs are able to convert A to I in RNA sequences, with no alterations in the genome. Furthermore, no specific PAM exists for the RBE modified from the Cas13 system. The I base is interpreted as G by the translational and splicing machinery, resulting in preferential base pairing with cytidine [67,68]. Abudayyeh et al. [29] expanded RESCUE as an efficient C-to-U RNA-editing tool by fusing dCas13 with cytidine deaminase. To make RESCUE even more efficient, rational mutagenesis of ADAR2DD was performed. Compared with DNA editing, RNA editing can be easily reversed and is limited to transcribed sequences [29]. Thus, RNA base editing is a promising technique that could be used for curing genetic diseases because it ensures safety, with no permanent changes occurring in genomic DNA [19]; however, it may be difficult to apply this tool for the improvement of plant agronomic traits.

## 3. Applications of Base Editing Tools for Plant Improvement

In plants, many genes have been edited using various BEs. As a result, precise mutations (such as an introduction of stop codons, amino acid changes, and regulatory site modifications) have been produced in target regions in the rice, wheat, maize, potato, watermelon, cotton, tomato, and *Arabidopsis* genomes [69,70] (Table 1). Compared with previous techniques, base editing is the most reliable tool for nucleotide substitution, being less time consuming and more precise [7]. Simplified steps for the base editing process are shown in Figure 3.

Different types of CBEs are also widely used in plant genome editing, particularly in rice and wheat. Several BEs, such as PBEs and hA3A-BE3, can efficiently perform base conversions in plants. A3A-PBE, generated from hA3A-BE3, has a large editing window (up to 17 bp), which can be used for mutagenesis-oriented editing [39]. Successful use of APOBEC3A (A3A-PBE) for the C-to-T base substitution was reported by Zong et al. [71], who showed that A3A-PBE was more efficient than was pnCas9-PBE, a standard PBE, enabling a conversion of C to T within a 7-nt editing window (from 3 to 9 bp). Additionally, pnCas9-PBE was generated via codon optimization of the nCas9-PBE fusion protein for cereal plants and cloned under the maize ubiquitin-1 gene promoter. Owing to the 17-nt editing window of A3A-PBE, the number of editable cytidines and guanidines is theoretically 1.8-fold higher than that of standard PBEs. In another study, two genes (*SLR1* and *NRT1.1B*), known to control the plant architecture and effective utilization of nutrients in rice, respectively, were targeted and edited via base editing, which resulted in increases in the height and nitrogen use efficiency in the mutant plants [20]. Moreover, by targeting four genes of rice (*OsAOS1*, *OsJAR1*, *OsJAR2*, and *OsCOI2*), Ren et al. [72,73] found that rice BE9 showed a higher editing efficiency and fewer unwanted mutations than did rice BE3, owing to the presence of the *UGI* gene in rBE9 (Table 1).

One of the most effective and promising applications of CBEs in plants has been development of herbicide resistance. Recently, an elite herbicide-resistant transgene-free wheat variety was developed by targeting the *TaALS-P174* gene, with an approximately 75% mutation efficiency, and the mutants showed high tolerance to imidazolinone, sulfonylurea, and aryloxyphenoxy propionate-type herbicides [38]. Acetolactate synthase (ALS) is a key enzyme of the biosynthesis of amino acids, and single-nucleotide substitutions at several positions in *ALS* genes have been shown to confer herbicide resistance to several plant species [70,83]. Zong et al. [69,71] also reported the development of herbicide resistance in wheat by targeting the *TaLOX2* and *ALS* genes using pnCas9-PBE and A3A-PBE. To date, base-editing systems have been successfully used not only for the modification of genes of herbicide resistance in rice and wheat but also for editing *ZmCENH3* and *ALS* genes in maize and watermelon [69,70]. Resistance to the herbicide tribenuron was induced in watermelons through a C-to-T conversion in the CCG codon (Pro190) in the *ALS* gene [70]. A3A-PBE, which can convert C to T within an editing window of 17 nucleotides, was extensively used for editing *StALS* and *StGBSS* genes in potato [71]. Additionally, transgene-free potato plants, with increased chlorsulfuron resistance, were produced by targeting the *ALS* gene using a CBE [82]. Moreover, the use of CBEs was shown to confer increased resistance to chlorsulfuron to tomatoes via the conversion of proline (CCA) to serine (TCA) in the targeted region of the *ALS* gene [82]. Marker-free tomato plants were also developed by editing *DELLA* and *ETR1* genes [53]. In another study, a CBE was reported with a window from 12 to 17 in the target region of cotton [78]. Base editing tools have also been applied in cotton by introducing a new BE (GhBE3) which was derived from the pRGEB32-GhU6.7 plasmid. Moreover, nCas9, UGI and a cytidine deaminase (APOBEC) was fused to generate pRGEB32-GhU6.7 plasmid. Using GhBE3, point mutations were introduced into the *GhCLA* (responsible for the chlorophyll content) and *GhPEBP* (controls branching of plants) genes, with mutation efficiencies of 26.67–57.78%. Taken together, these findings demonstrated that base-editing tools might be effectively used for editing of target genes and might have broad applications for the improvement of plant traits.

Notable findings were reported in studies developing specific traits in plants using evolved BEs (from useful BEs to effective PBEs), and the efficiencies of these BEs were also evaluated. Li et al. [74] reported that the ABE system could be used to efficiently introduce point mutations in plants. Six rice genes (*ACC*, *ALS*, *CDC48*, *DEP1*, *NRT1.1B*, and *OsEV*) were edited using a successfully developed ABE [74]. Furthermore, PABE-7 BE, which exhibits high mutation efficiency, has been widely used. Hua et al. [84] evaluated the effectiveness of multiple ABEs using *IPA1* (*OsSPL14*), *OsSPL17*, *OsSPL18*, and *SLR1* genes. Two ABEs, ABE7.8 and ABE7.10, were used to edit *MPK6*, *MPK13*, *SERK2*, *WRKY45*, and *Tms9-1* genes [75,76], suggesting that base editing tools can be used for crop improvement. The mutation efficiency of ABEs was higher than that of CBEs, based on the editing of the *OsACC*, *OsALS*, *OsDEP1*, *OsNRT1*, *OsCDC48*, and *OsWx* genes in rice [42]. Similar conclusions have been reported in other studies [40,47,85]. Furthermore, some studies of mutation efficiencies of ABEs demonstrated that unwanted mutations were less frequent with ABEs than when using CRISPR/Cas9 tools [86]. Liang et al. [40] developed EndoV-seq, utilizing endonuclease V to nick an inosine-containing strand of genomic DNA, which was deaminated by ABEs in vitro. Using EndoV-seq, the authors evaluated genome-wide off-target deamination by ABEs and then screened and scored six different gRNAs in a single EndoV-seq assay. They suggested the specificity of ABE might be improved through selective modification of gRNA length without sacrificing its on-target efficiency, and also found that ABE induced less off-target mutations than classical CRISPR-Cas9 system [40]. Using ABE7.10, another study has expanded base editing in rice and wheat by targeting *DEP1*, *TaEPSPS*, and *GW2* genes [74]. Four *Arabidopsis* genes (*AtALS*, *AtPDS*, *AtFT*, and *AtLFY*) and two rapeseed genes (*BnALS* and *BnPDS*) were edited with the same strategy [81]. As a result, *AtFT*-knockout transgenic plants were found to exhibit delayed flowering, whereas *AtPDS3*-knockout transgenic plants were found to exhibit dwarfism and a mosaic albino phenotype using ABE7.10 in *Arabidopsis* [81]. Thus, base editing tools can be effectively used to improve agronomic traits in plants.

## 4. Future Perspectives and Limitations

Although base-editing systems, with their simplicity and precision, have rapidly advanced genome editing, there are still many aspects of this technology that should be explored [22] and many challenges that should be overcome before these systems become reliable tools for genome editing. First, the currently available PBEs can only change purines to purines and pyrimidines to pyrimidines [22,26]. Thus, it is necessary to develop BEs that can change purines to pyrimidines and vice versa in order to facilitate a wider application of BEs in plants. Recently, Anzalone et al. [87] have developed a search-and-replace genome-editing technology as a prime editing tool by fusing a Cas9 nickase with a reverse transcriptase to mediate all possible base conversions in human cells. Efficient base transversions and small indels were successfully obtained in human cells and mouse neurons using prime editors [87,88]. Recently, Lin et al. [89] optimizes prime editing in plants promoter, codon, and editing-condition optimization. Prime editing is less efficient than BEs for making transition point mutations; although prime editors can generate substitutions, insertions, deletions, and transversions. Thus, this new technology expands the scope of genome editing. Cpf1-based CBEs, which perform the conversion of C to T but are limited by the presence of T-rich PAM sequences, have successfully been used in human cells [57] and in addition to C-to-T mutations, some currently available CBEs have been shown to mediate C-to-G and C-to-A substitutions [53,82]. Base editing is more PAM flexible, as various modified or engineered Cas9 proteins have been used in BEs [39,47,50]. Thus, to make base editing tools more precise, reliable, and durable, it is necessary to increase their compatibility with PAM sequences.

The second challenge is off-target mutations, which are considered a major limitation of genome editing systems. Off-target activity of BEs or Cas9-linked deaminases is still not clear. Kim et al. [84] reported off-target effects of BEs and Cas9 as independent assessment of their genome-wide specificities, suggesting that a separate off-target mutation evaluation method may be needed for base editing. Previous studies have indicated that CBEs contain more off-target mutations than ABEs [40,42,47,85]. To overcome this, Doman et al. [54] engineered CBE variant that maintain minimum Cas9-independent off-target editing. They concluded CBEs with YE1 deaminase domain enable high-fidelity base editing. Precise CBEs with low off-target mutation also reported by engineering CDA1 truncations and nCas9 fusions [43]. Recently, Li et al. [90] engineered dual BEs as saturated targeted endogenous mutagenesis editors to generate mutation in plants. They fused adenine deaminase with cytosine deaminase to create dual adenine and cytosine BEs. Hopefully, these improved BEs will be widely applicable in near future. Therefore, to overcome the APOBEC-mediated cytidine deamination-dependent U:G mismatches, Wang et al. [91] developed an enhanced BE by co-expressing free UGI with BE3 to minimize unwanted mutations. Rees et al. [46] successfully mutated BE3 to a high-fidelity BE to reduce off-target C to T conversion. Recently improved CBEs with less DNA affinity showed a higher editing efficiency [41,42]; and selective curbing of unwanted RNA editing (SECURE)-BE3 variants were developed to reduce off-target RNA editing activity [92]. Indeed, replacement of APOBEC1 with a human APOBEC3A resulted in an increased efficiency of RNA base editing using CBEs. Researchers also developed SECURE-ABE variants with reduced off-target RNA-editing efficiency using miniABAmax, which lacks the wild-type TadA domain from ABEmax. Taken together, these results suggest that there is large developmental potential for increasing the mutation efficiency of BEs in the future.

Furthermore, ABEs can generate efficient mutations and alter *Arabidopsis* agronomic traits, such as flowering time, plant height, pathogen resistance and albino phenotype by editing *AtFT*, *AtPDS3,* and *AteIF4E* genes [80,81]. BEs were also found to improve plant architecture and nitrogen uptake efficiency in rice [20]. Both ABEs and CBEs can play essential roles in the improvement of plant architecture, resistance, and nutrient uptake efficiency [38,53,69,81]. Working with genes conferring resistance is effective for the development of genome editing-tools in the lab [93]. However, producing herbicide-resistant crops for commercial application need future concern for the development of a sustainable agriculture and for public acceptance of genetic engineering. Wider applications of base editing are necessary to make base editing more reliable. Previously developed RBE systems can convert A to I, and Abudayyeh et al. [30] recently reported a RESCUE approach which works as a C to U RNA editor. However, additional studies are needed to develop efficient RBE systems for C to U substitution [29], and the stability of RBE tools has not yet been reported in plants. 

Finally, BEs can target either C or A within the editing window; however, it remains unknown which base will be converted within an editing window. Generally, BEs with larger editing windows, such as A3A-PBE, are desirable because of a large number of editable bases [71]. However, with regard to precision, BEs with narrow editing windows are also necessary, owing to a limited specificity of those with large editing windows. Tan et al. [56] have reported a highly precise CBE for single-nucleotide substitutions. PBEs with a single-nucleotide editing window could be a key for achieving high editing efficiency with more specific targets.

Base editing is a recently developed and advanced technology. However, there is still a need for developing easier, more precise, and more convenient base-editing tools. Base-editing tools have been developed within a shorter period of time than were other CRISPR-based tools. Thus, more applications of base-editing systems in crop improvement are expected in the near future.

## Figures and Tables

**Figure 1 genes-11-00466-f001:**
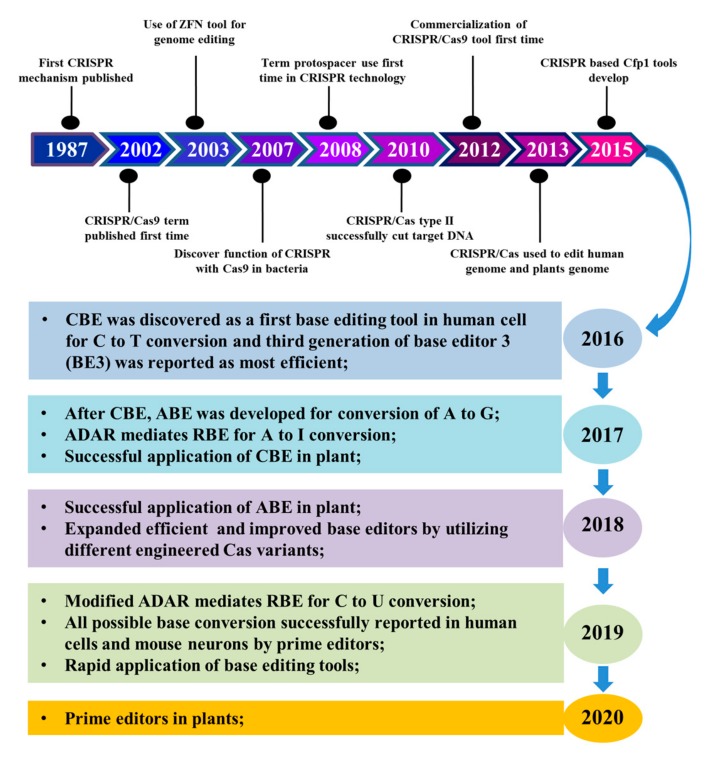
**Progress of genome editing.** Milestones of genome editing tools and brief history of base editing. ZFN, zinc-finger nucleases; CBE, cytosine base editor; ABE, adenine base editor; ADAR, adenosine deaminase acting on RNA; RBE, RNA base editor; CRISPR, Clustered regularly interspaced short palindromic repeats; Cas9, CRISPR associated protein 9.

**Figure 2 genes-11-00466-f002:**
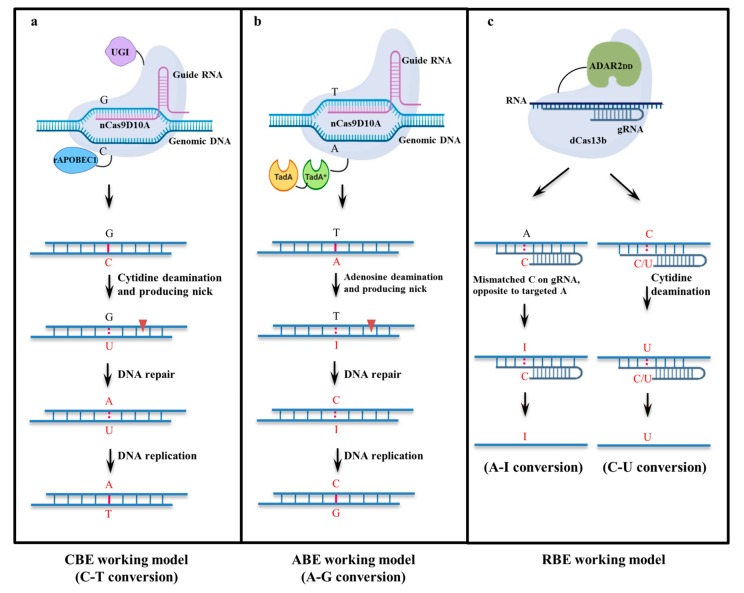
**Base editing mechanisms.** (**a**) Cytosine base editor: C to T base editing strategy; (**b**) Adenine base editor: A to G base editing strategy; (**c**) RNA base editing mechanisms: A to I (left) and C to U (right) RNA base editing strategies. Abbreviations: UGI, uracil glycosylase inhibitor; rAPOBEC1, rat cytidine deaminase; nCas9, a DNA nickase; ABE, adenine base editor; TadA-TadA* (TadA, wild-type *Escherichia coli* transfer RNA (tRNA) adenosine deaminase; TadA*, mutated TadA); dCas13, catalytically inactive Cas13; ADAR2DD, adenosine deaminase acting on RNA.

**Figure 3 genes-11-00466-f003:**
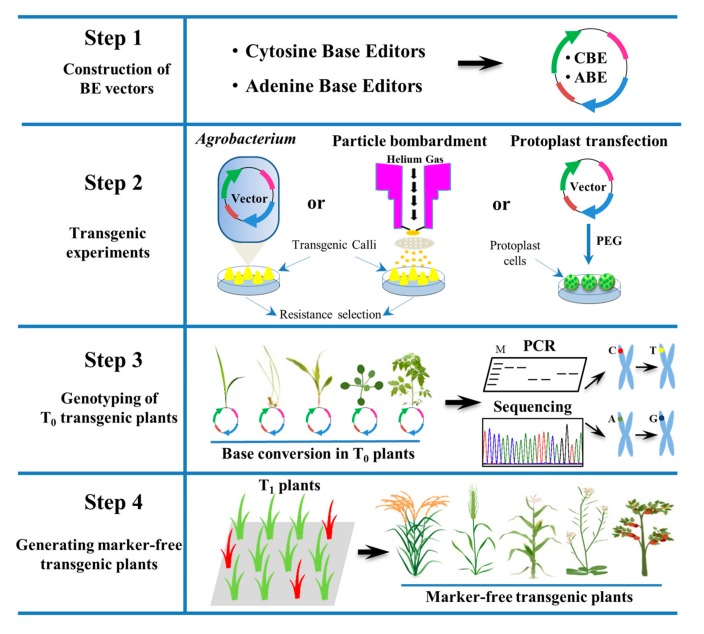
**General steps of base editing**. Steps of base editing tools: Construction of vectors, transformation, and mutants screening. According to Mendelian segregation laws, 25% plants will be marker-free transgenic plants in T_1_ generation. Base editing successfully improves different traits of rice, wheat, maize, *Arabidopsis*, tomato and other crops.

**Table 1 genes-11-00466-t001:** Application of base editing in different plant species.

Plant Species	Targeted Gene	Selected PAM	Base Editor	Mutation Efficiency	Editing Window (nt)	Improved Trait or Key Findings	Reference
Rice^1^	*SLR1* *NRT1.1B*	AGG GGG	APOBEC1-XTEN-Cas9(D10A)	13.3% 2.7%	4 to 8	Reduced plant height; increased nitrogen use efficiency	[20]
Rice^1^	*ACC*, *ALS*, *CDC48*, *DEP1, NRT1.1B* *OsEV*	CCT	ABE7.10	3.2–59.1%	4 to 8	Development of efficient ABE PABE-7	[74]
Rice^1^	*ALS,* *FTIP1e*	AGG CCA	Target-AID	6–89%	−19 to −17	Develop multiple herbicide resistance	[53]
Rice^1^	*OsAOS1* *OsJAR1* *OsJAR2* *OsCOI2*	CCA TGG	rBE3 rBE9	8.3–73.3%	−19 to −13	Prove editing efficiency of rBE9, which is higher than rBE3	[73]
Rice^1^	*OsCERK1* *OsSERK1* *OsSERK2* *ipal* *pi-ta* *BRI1*	NGA AGTG AGCG	rBE3	10.5–38.9%	−19 to −13	Detect the efficiency of rBE3	[72]
Rice^1^	*IPA1 (OsSPL14)* *OsSPL17* *OsSPL18* *SLR1*	GAG CAG CGA GGA AGCG GGCG	ABE-P1 ABE-P2 ABE-P3 ABE-P4 ABE-P5	26%	3 to 15	Multiple adenine base editor evaluation	[50]
Rice callus^1^	*sgOs-siteG1* *sgOs-site2* *sgOs-site3* *sgOs-site4*	NGG NGA NGC NGT	ABE7.10	29.2–45.8%	13 to 16	Develop new ABEs	[75]
Rice^1^	*OsCDC48*, *OsNRT1.1B* *OsSPL14*	CGG	pnCas9-PBE	43.48%	3 to 9	Reduce senescence and death	[69]
Rice^1^	*OsNRT1.1B* *OsCDC48*	NGG CCN	A3A-PBE	44.1% 82.9%	1 to 17	A3A-PBE editor is more efficient than pnCas9-PBE	[71]
Rice^1^	*EPSPS*, *ALS*, *DL*	NG	Target-AID-NG	5–95.5%	−9 to −20	SpCas9-NGv1 application in base editing	[49]
Rice^1^	*MPK6*, *MPK13*, *SERK2*, *WRKY45*, *Tms9-1*	CCA CCG	ABE7.10 ABE7.8	0–62.26%	−17 to −11	Develop new adenine base editor using fluorescence-tracking	[76]
Rice^1^	*OsACC* *OsALS* *OsDEP1* *OsNRT1* *OsCDC48* *OsWx*	AGG TGG CCA CCT CGG GGG	Be3 HF1-BE3 ABE(PABE-7)			Off-target mutation is higher in CBE compared to ABE.	[42]
Rice^1^	*GL1-1* *NAL1*		nCas9-PBE	58% 68%	3 to 9	The mutant with hydrophilic leaf surface and abnormal transcripts of *NAL1*	[77]
Wheat^2^	*TaLOX2*	CGG	pnCas9-PBE	1.25%	3 to 9	Herbicide resistance	[69]
Wheat^2^	*TaALS-P174*	CGG CCT	PBE	33–75%	3 to 9	Increase multiple herbicide resistance	[38]
Wheat^2^	*DEP1*, *TaEPSPS GW2*	CCT	ABE7.10	0.4–1.1%	4 to 8	Increase herbicide resistance	[74]
Wheat^2^	*ALS gene*	NGG CCN	A3A-PBE	16.7–22.5%	1 to 17	Herbicide resistance and editing efficiency of A3A-PBE	[71]
Maize^1^	*ZmCENH3*	CGG	pnCas9-PBE	10%	3 to 9	Bialaphos-resistant	[69]
Cotton	*GhCLA* *GhPEBP*	TGG CCA AGG	*G. hirsutum*-Base Editor 3 (GhBE3)	26.67–57.78%	−17 to −12	Point mutation was generated with novel GhBE3 in cotton	[78]
Watermelon^1^	*ALS gene*	TGG CGG	BE3	23%	3 to 9	Herbicides resistance	[70]
*Arabidopsis* ^1^	*ALS gene*	TGG	BE3	2.7–40%	4 to 9	Inheritable herbicides resistance was found	[79]
*Arabidopsis* ^1^	*eIF4E1*	NGG	CBE	50%		C-to-G base editing generate Clover yellow vein virus resistant plants	[80]
*Arabidopsis* ^1^	*AtALS* *AtPDS* *AtFT* *AtLFY*	TGG AGG GGG CGG	ABE7.10 (pcABE7.10)	0–85%	1 to 12	Plant ABE application	[81]
Tomato^1^	*DELLA* *ETR1*	AGG CCA	Target-AID	41–92%	−19 to −17	Generate marker-free plants	[53]
Tomato^1^	*A* *LS*	TGG	CBE	71%	−20 to −13	Obtain of Chlorsulfuron-resistant	[82]
Potato^3^	*StALS StGBSS*	NGG CCN	A3A-PBE	6.5%	1 to 17	Widespread use of A3A-PBE in dicotyledons	[71]
Potato^1^	*ALS*	TGG	CBE	100%	−20 to −13	Herbicide resistant	[82]
Rapeseed^1^	*BnALS BnPDS*	TGG AGG GGG CGG	ABE7.10 (pcABE7.10)	8.8%	1 to 12	Plant ABE application	[81]

Abbreviations: ^1^
*Agrobacterium* mediated system; ^2^ Particle bombardment; ^3^ Protoplast transfection.

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
