# Peer review of "Base Editing: The Ever Expanding Clustered Regularly Interspaced Short Palindromic Repeats (CRISPR) Tool Kit for Precise Genome Editing in Plants"

_genes, 2020, doi:10.3390/genes11040466_

Round 1

Reviewer 1 Report

The manuscript entitled “Base editor: the ever expanding CRISPR toolkit for precise genome editing in plants” from Monsur et al. aimed to provide up-to-date information about the development of base editor tools for plants. Although the authors relied on relevant studies and examples from this field, the manuscript suffers from some mistakes and is overall confusing for the reader. Some ideas are not illustrated by the right examples and there is a lot of redundancy throughout the manuscript. Furthermore, recent studies providing great tools for precise and efficient base editing have not been described, and therefore should be discussed (Li et al., 2020; Tan et al., 2020; Doman et al., 2020).

Some major points:

- Line 43-45: Why do the authors claim that classical CRISPR/Cas9 system is questionable for mutation frequency and off-target effects? Several studies reported a high mutation efficiency with low off-target effects in many plant species…

- Lines 47-48: HR in plants is not only limited by the availability of donor templates into plant cells (Huang and Puchta, 2019)

- Line 52: The Cas protein is not inactive but impaired for DNA cutting

- Line 54-56: Base editing is not more efficient than other methods due to the lack of DSB…

- Lines 79-88: This section is very confusing. The authors claim that “the rate of off-target mutations is lower in precise base editing than CRISPR/Cas9”. Recent work in rice demonstrated that CBEs induce genome wide off-target mutations (Jin et al, 2019), while recent studies showed that these unwanted effects can be strongly limited (Tan et al, 2020; Doman et al, 2020). This should be discussed in more details.

- Table 1: The authors compare different CRISPR systems (Cas9, cpf1, Cas13) to the base editing technology. As BE characteristics are highly dependent of the Cas protein, I find this comparison not relevant. Furthermore, Prime editing is not a base editing technology.

- Line 101: uracil DNA glycosylase

- Lines 115-116: Cas9 variants do not alter the PAM, they recognize different PAMs.

- Lines 124-125: The study by Qin et al (2019) should be discussed about the use of SaCas9 and SaCas9KKH for base editing in plants

- Line 130: recognize instead of “create”

- Figure 2: Deamination occurs on the non-targeted strand, therefore the figure should be corrected for better understanding.

- Lines 152-154: xCas9 does not work efficiently in plants. The authors should take into account several studies using the Cas9-NG variant (Ge et al, 2019; Hua et al, 2019; Zhong et al, 2019; Ren et al, 2019; Niu et al, 2019; Li et al, 2020; Veillet et al, 2020).

- Figure 3: Protoplast transfection should be included

- Table 2: Please include spaces between examples

- Lines 204-205: Working with genes conferring herbicide resistance is effective for the development of genome editing tools in the lab. However, producing herbicide-resistant crops for commercial application may not be a promising track for the development of a sustainable agriculture and for public acceptance of genetic engineering.

- Lines 259-260: in addition to C-to-T mutations, some currently available CBEs have been shown to mediate C-to-G and C-to-A substitutions (Shimatani et al, 2017; Veillet et al, 2019).

- Lines 266-271: is Cpf1 a good candidate for base editing in plants? To my knowledge, no Cpf1 nickase is available, strongly limiting the use of Cpf1 for base editing. Please provide information about the importance of nicking the non-edited DNA strand and the relevance of cpf1 for base editing.  

- Lines 276-277: I though the deaminase, not the nuclease, was responsible for genome-wide off-target? Please clarify this statement. Please also include the recent study performed by Doman et al (2020).

- Lines 288-293: Base editing has also been used for pathogen resistance (Bastet et al, 2019).

- Lines 298-305: Please include the study from Tan et al. (2020) for high precision base editing and the study from Li et al. (2020) for the use of dual cytosine and adenine base editors.

- The authors only discuss about APOBEC enzymes for CBE. However the PmCDA1 enzyme was also broadly used by the plant science community. Therefore, a comparison between APOBEC enzymes and PmCDA1 should be included.

- The manuscript should be carefully corrected for English language

Reviewer 2 Report

“Base editing:  the ever expanding CRISPR took kit…….in Plant” by Monsur et al.

A timely review that efficiently summarizes the current status of a fascinating and fast-moving topic in biology in general, including plant biology.

For the most part, the review is well-written.  However, the biggest flaw is the profuse misuse of the terms “precise (or precision)”, “reliable”, “efficient (or efficiency)”.  These three terms, and derivations of them, are thrown around somewhat randomly throughout the manuscript.  Often they are used incorrectly. Each of these terms have very specific definitions, and the proper use of them, as they pertain to explaining the relative strengths and weaknesses of different CRISPR base editing systems, REALLY MATTER.  Look up the definitions of these words in the dictionary, then go back and carefully review the manuscript, stopping at each point where these words come up and ask yourselves which term truly best describes the enzyme property that you are discussing.

Likewise, be very specific about the use of “substitutions” versus “conversions”.  If you are using a base editor, and it changes a C to a T, the Cas enzyme DID NOT perform a substitution.  Substitution implies that a base was removed then replaced with a different base.  BE’s don’t do that.  They convert one base to another, all the while the backbone of that base remains in place in the DNA (or RNA) strand. 

Words matter.  Precise meanings matter.  Do better, and clean this up.  Otherwise, this manuscript was useful and will be a good resource for people hoping to get a thorough overview of base editing.

Round 2

Reviewer 1 Report

The manuscript has been improved compared to the first version. Nevertheless, there are still some mistakes and redundancy.

-l12: has

-l14-16: PAM preference limitaion has been overcome by Cas variants, not base editing tools

-l40: NHEJ is also efficient and widely used for single gene editing

-l45-46: A recent study has been published during the reviewing process, with great potential: Ali et al, 2020: Fusion of the Cas9 endonuclease and the VirD2 relaxase facilitates homology-directed repair for precise genome engineering in rice

-l81-82: Are you refering to gene targeting through CRISPR-Cas9? 

-l84-85: In Jin et al (2019), CBE induced more base conversion than the control and ABE, but not indels

-l85-89: this section has been strongly improved

-l94-96: There are some unwanted mutations (non C-to-T conversion, indels, bystander mutations), especially with early CBE

More efficient compared to what? Gene targeting?

-l113-114: "Zong et al. [47] fused human APOBEC3A with a Cas9 nickase and produced an effective PBE, A3A-PBE, with a 17-nucleotide editing window." This sentance is not related to this section about Cas variants 

-l117-120: Please precise the SaCas9 PAM, which is mainly NNGRRT

-l123-125: It may be better to add the sentences about SaCas9 in plants (l117-120) after this example.

-l125-126: Please cite Shimatani et al (2017) for the development of PmCDA1 for CBE in plants

-l131-139: this section has been strongly improved

-Figure 2: Please could you indicate the cutting site of nCas9 for CBE and ABE in order to stimulate DNA repair mechanisms using the edited strand?

-l157-159: Please cite the articles describing xCas9 (Hu et al, 2018) and SpCas9-NG (Nishimasu et al, 2018)

-Figure 3: Thanks for adding the protoplast transfection. However, there is no selection for most applications using this method

-l227-228: I do not understand how a single CBE could have six editing windows. Please clarify this sentence

-l230-231: I do not understand why the selection marker facilitate sgRNA transcription...

-249-253: In Liang et al, the authors assessed the off-target activity, and found that ABE induced less off-target mutations than classical CRISPR-Cas9 system. However, I did not find data about the efficiency at on-target sites. Maybe the authors refer to "precise" rather than "efficiency"?

-l302-303: This is wrong and contradictory with next sentences.

-l304: in Bastet et al, pathogen resistance was confered by edition of the AteIF4E gene
